# Current Advances on the Important Roles of Enhancer RNAs in Molecular Pathways of Cancer

**DOI:** 10.3390/ijms22115640

**Published:** 2021-05-26

**Authors:** Rui Wang, Qianzi Tang

**Affiliations:** Institute of Animal Genetics and Breeding, College of Animal Science and Technology, Sichuan Agricultural University, Chengdu 611130, China; ryan.wang.sicau@gmail.com

**Keywords:** eRNA, cancer signaling pathway, cancer therapy

## Abstract

Enhancers are critical genomic elements that can cooperate with promoters to regulate gene transcription in both normal and cancer cells. Recent studies reveal that enhancer regions are transcribed to produce a class of noncoding RNAs referred to as enhancer RNAs (eRNAs). Emerging evidence shows that eRNAs play important roles in enhancer activation and enhancer-driven gene regulation, and the expression of eRNAs may be a critical factor in tumorigenesis. The important roles of eRNAs in cancer signaling pathways are also gradually unveiled, providing a new insight into cancer therapy. Here, we review the roles of eRNAs in regulating cancer signaling pathways and discuss the potential of eRNA-targeted therapy for human cancers.

## 1. Introduction

Enhancers are regions of DNA that drive transcription independently of their distance and orientation from the gene, in contrast to promoters which exert their functions in an orientation-dependent manner. Enhancers contain binding sites for RNA polymerase (RNA pol) II, multiple transcription factors, and co-regulators, and can interact with promoters to form spatial chromatin loops. As critical genomic elements, enhancers regulate gene transcription in various diseases including human cancers, and enhancer malfunctions can have a direct effect on tumor growth.

Enhancer regions themselves can also be actively transcribed and produce a type of non-coding RNA termed as enhancer RNAs (eRNAs), which were first detected in 2010 [1]. eRNAs are nascent RNA transcripts that exhibit a 5′ cap, and they were first observed to be short bidirectionally transcribed RNA transcripts of no more than 2 kb and were non-polyadenylated [1]. Soon after, it was discovered that there were also polyadenylated eRNAs, and interestingly, these polyadenylated eRNAs were generally unidirectionally transcribed and longer than those that were non-polyadenylated. However, according to FANTOM5 enhancer atlas [2], bidirectionally transcribed, non-polyadenylated eRNAs are more common, and the majority of eRNAs are not spliced. The process of eRNA transcription is triggered by the binding of specific transcription factors and coactivators to enhancers. Then, they recruit other transcription factors, complex and histone modifiers such as P300/CBP. H3K27 is acetylated by P300/CBP and the enhancer regions are further opened, leading to the recruitment of essential proteins such as RNA pol II. Cofactors such as BRD4 promote RNA pol II elongation and eRNA processing [3]. Although eRNAs are commonly observed, their roles are largely unknown, and it has been suggested that they are a byproduct of transcription. However, a growing number of studies have indicated diverse roles for eRNAs in regulating gene transcription and many other aspects of cell functions. Here, we review the roles of eRNAs in cancer gene expression and signaling pathways and discuss the potential of eRNA-targeted therapy for human cancers.

eRNAs define active enhancers and co-localize with epigenetic markers, and it is widely accepted that eRNA production is closely associated with enhancer activation. Indeed, the expression of eRNAs positively correlates with the level of histone H3 acetylation on lysine 27 (H3K27ac), which is a marker for active enhancers [4]. Moreover, eRNA transcription is reported to be a better marker than H3K27ac for identifying enhancer activation and is shown to forecast novel enhancers [5,6,7].

Super enhancers (SEs) consist of large clusters of transcriptional enhancers that activate cell type- and tissue-specific genes. Compared with typical enhancers, SEs have specific features such as a significantly increased transcription factor occupancy, high levels of H3K4me1 and H3K27ac density, and increased occupancy of RNA pol II, mediator, p300/CBP, and bromodomain-containing (BRD)4 cofactors. SE transcription also produces higher quantities of eRNAs [8,9,10,11,12].

eRNA transcription via RNA pol II precedes the activation of the target gene, suggesting that eRNAs are regulators of enhancer-mediated target gene expression [7,13,14,15,16]. Indeed, the production of eRNAs is positively correlated with the expression of the target genes of corresponding enhancers [1,2,3,17], with eRNA knockdown typically reducing target gene expression. For example, eRNA production from p53-bound enhancer regions (p53BERs) is required for the p53-dependent activation of gene expression, while the knockdown of p53BER2- and p53BER4-derived eRNAs inhibit the induction of genes closest to these two p53BER domains [18].

eRNAs appear to play key roles in binding transcription factors needed for the up-regulation of enhancer-mediated genes; for example, colon cancer-associated transcript (CCAT) 1 recruited transcription factors TP63 and SOX2 to SE regions of the epidermal growth factor receptor gene (EGFR) to promote its transcription [19]. Similarly, interactions have been reported between eRNAs and specific transcription factors, such as BRD4, P53, and cohesin, which bind a broad array of enhancers to promote gene transcription [11,15,20]. Although enhancers are cis-acting regulatory elements located some distance from transcription start sites, they still effectively promote the expression of target genes independently of their position and orientation because they can loop over long genomic ranges to engage distant promoters. Previous studies suggested that eRNAs might promote enhancer–promoter looping and recruit transcription factors to specific enhancers [8,21,22,23] (Figure 1).

Indeed, eRNAs were reported to act mostly in-cis to promote enhancer–promoter looping. For example, eRNAs transcribed from enhancers adjacent to 17β-estradiol up-regulated genes play important roles in their induction by strengthening enhancer–promoter looping through estrogen receptor α binding [20]. Similarly, eRNAs produced from an enhancer located upstream of the kallikrein-related peptidase 3 gene (KLK3) promote the interaction of the KLK3 enhancer and KLK2 promoter, thus enhancing the long-distance transcriptional activation of KLK2 [13]. Furthermore, Epstein Barr virus (EBV) SE eRNAs are important for enhancer–promoter looping at the MYC locus, while eRNA CCAT1 interacts with CTCF to promote chromatin looping between the MYC promoter and enhancers, leading to MYC gene expression [24]. As such, eRNA knockdown can disrupt the loop structures and decrease target gene transcription [13,24,25,26].

## 2. eRNA in Oncogene Expression Regulation and Tumorigenesis

Although the eRNA molecular mechanisms in human cancers are not yet fully understood, abundant evidence suggests that the abnormal expression of eRNAs is closely related to tumorigenesis. Zhang et al. examined differentially expressed eRNAs in matched tumor-normal samples across 16 cancer types, and found more up-regulated eRNAs (median, 42.2%) than down-regulated ones (median, 9.9%) [27]. Similarly, Qin et al. observed elevated global eRNA expression among lung adenocarcinoma tumor samples compared with control samples [28]. It has also been reported that individual eRNAs play important roles in tumorigenesis in many cancer types (including lung adenocarcinoma, prostate cancer, breast cancer, and squamous cell carcinomas), mainly through regulating carcinoma-related genes.

In lung adenocarcinoma, Qin et al. identified hundreds of eRNAs to be functional as their correlated genes were over-represented in cancer driver and clinically actionable gene ensemble. The eRNA located upstream of telomerase reverse transcriptase (TERT), a well-known predisposition gene for lung cancer [29], appears to contribute to cancer development by up-regulating TERT expression. Moreover, the copy number amplification of Forkhead box (FOX)O6 eRNA contributes to the up-regulation of FOXO6 in lung adenocarcinoma [28].

Similarly, a group of androgen receptor (AR)-regulated eRNAs, including prostate-specific antigen eRNA (PSA eRNA), were found to be up-regulated in castration-resistant prostate cancer (CRPC) cells, patient-derived xenografts, and patient tissues. These AR-regulated eRNAs have important roles in the regulation of AR target genes. For example, PSA eRNA bound cyclin T1 and activated the positive transcription elongation factor, then increased serine-2 phosphorylation of RNA pol II (Pol II-Ser2p), which is essential for transcription elongation and gene expression. Additionally, PSA eRNA knockdown significantly decreased Pol II-Ser2p levels at the loci of a subset of genes, including those involved in regulating the cell cycle, growth, survival, migration, and invasion, such as VEGFA, NCAPD3, ADAMTS1, and IGF1R. An HIV-1 trans-activation response element RNA-like motif in PSA eRNA was also shown to be essential for increased Pol II-Ser2p occupancy levels and CRPC cell growth [30].

The high expression of NET1e, an eRNA located about 90 kb downstream of the oncogene NET1, was detected in breast cancer, and NET1e knockdown significantly reduced the proliferation of the breast cancer cell line MCF7. NET1e appears to contribute to breast cancer progression by up-regulating NET1 expression [27].

Finally, the knockdown of oncogenic eRNA CCAT1 in squamous cell carcinomas cells down-regulates the expression of SE-associated genes involved in cell proliferation, growth, and migration, as well as those with a role in DNA-protein interactions. This suggests that CCAT1 functions in cancer progression and the interaction of macromolecules in cancer [19].

Taken together, these results suggest important roles for eRNAs in various cancer types.

## 3. eRNA-Associated Signaling Pathways

Besides regulating carcinoma-related genes, eRNAs also associate with signaling pathways in different kinds of cancers. Zhang et al. identified differentially expressed eRNAs between tumor and control samples in prostate cancer, of which, 295 were correlated with the expression of 325 target genes [31]. Functional enrichment analysis showed that these target genes were over-represented in 20 Kyoto Encyclopedia of Genes and Genomes pathways related to inflammation, cellular adhesion, and apoptosis. GeneAnalytics analysis showed that 17 of these target genes participated in pathways related to prostate cancer development, such as phosphatidylinositol-3-kinase (PI3K)–Akt signaling, AMP-activated protein kinase signaling, focal adhesion, regulation of the actin cytoskeleton, extracellular matrix–receptor interactions, and proteoglycans in cancer [31].

Further effects of eRNAs in signaling pathways were shown by Zhang et al. [27]. They collected 229 genes in 10 cancer signaling pathways (the cell cycle, Hippo, Myc, Notch, Nrf2, PI3K, receptor tyrosine kinase [RTK]-RAS, transforming growth factor-β, p53, and Wnt), and found that 80.8% of these genes were correlated with eRNAs in at least one cancer type. These genes included those in the p53 pathway (such as MDM2, MDM4, ATM, CHEK2, RPS6KA3, and TP53), the PI3K signaling pathway (such as PTEN, AKT2, AKT1S1, RPS6, and PIK3R3), the Notch signaling pathway (such as KDM5A, DLK1, DLL3, NOTCH1, and NOTCH2), and the RTK–RAS signaling pathway (such as IRS2, JAK2, and KSR2) [27].

This genomic and computational evidence suggests an association between eRNAs and various signaling pathways in cancers, which is further supported by experimental evidence. For example, the oncogenic eRNA CCAT1 plays important roles in activating both MEK/ERK1/2 and PI3K/AKT signaling pathways [19].

ERK1/2 are conserved kinases that regulate cell signaling and are critical members of the mitogen-activated protein kinase (MAPK)/ERK signaling pathway in cancer development. The MAPK signaling pathway regulates a wide range of cellular functions, such as differentiation, proliferation, apoptosis, and stress responses [32,33]. The PI3K signaling pathway plays a crucial role in regulating diverse cellular functions, including metabolism, growth, proliferation, survival, transcription, and protein synthesis. It is one of the most frequently dysregulated pathways in human cancers that is activated in a wide spectrum of cancers, such as thyroid cancer, nasopharyngeal cancer, lung cancer, esophageal cancer, colorectal cancer, brain cancer, and others. AKT is a serine protein kinase that links to PI3K activity in all cell types [34,35,36].

Jiang et al. showed that transcription factors TP63 and SOX2 regulate CCAT1 expression in squamous cell carcinoma cells, and that CCAT1 in turn recruits and forms a complex with TP63 and SOX2 [19]. This complex binds to and induces the EGFR SE, which further activates MEK/ERK1/2 and PI3K signaling pathways (Figure 2A). Using Western blotting and the human phospho-kinase array, they observed CCAT1 silencing in squamous cell carcinoma cells and xenografts, and reduced the phosphorylation of critical molecules mediating the two signaling pathways compared with controls [19].

As well as being involved in regulating signaling pathways, eRNAs are also in turn regulated by pathways, including nuclear factor (NF)-κB, PI3K, and MAPK/ERK.

NF-κB is part of a family of transcription factors that play important roles in cell survival, cytokine production, DNA transcription, and other cellular events [37,38]. Family members include p65 (RelA), RelB, c-Rel, NF-κB1, and NF-κB2 [39], while the heterodimer of p50 and p65 is the most extensively studied form of NF-κB. NF-κB can be activated by a variety of stimulants, such as lipopolysaccharide (LPS), inflammatory stimuli, cytokines, tumor promoters, and two kinase-dependent pathways: the canonical and non-canonical NF-κB pathway. In the former, the inhibitor of κB kinase (IKK)β, but not IKKα, is required for NF-κB activation; the latter is dependent on IKKα homodimers rather than IKKβ or IKKγ [40,41].

A study in 2015 measured the expression of a subset of eRNAs that are induced in response to LPS stimulation of monocytic THP-1 cells, including SLC30A4-eRNA, SOC3-eRNA, AZIN1-eRNA, TNFSF8-eRNA, MARCKS-eRNA, and ACSL1-eRNA [42]. LPS-induced eRNA expression was significantly reduced following the inhibition of IKK2, an upstream activator of NF-κB. Small interfering (si) RNA-mediated knockdown of p65 and chromatin immunoprecipitation combined with real-time quantitative PCR revealed the significantly reduced expression of some but not all LPS-induced eRNAs, including AZIN1-eRNA, TNFSF8-eRNA, MARKS-eRNA, and ACSL1-eRNA. While the NF-kB signaling pathway appeared to regulate these LPS-induced eRNAs, AZIN1-eRNA and TNFSF8-eRNA were also controlled by ERK-1/2 pathways and the p38 pathway, respectively (Figure 2B) [42,43].

## 4. eRNA and Cancer Therapy

eRNAs are functional molecules that modulate gene expression, such as that of important signaling pathways and immune checkpoints, in various cancers. Therefore, there is the potential for direct eRNA-targeted therapy in human cancers, and eRNAs could be used as clinical markers for predicting and monitoring therapeutic responsiveness and resistance.

It is reported that three different eRNAs are associated with distinct cancers, including prostate cancer-associated transcript 1 with expression related to prostate cancer, LOC728724 related to T cell acute lymphoblastic leukemia, and CCAT1 related to colon cancer. It is observed that bromodomain and extra-terminal (BET) inhibition led to near-complete RNA reduction for each of these eRNAs [21,44]. As cMYC is an established BET target and the expression of these eRNAs is associated with BET-mediated cMYC transcription [45], the eRNAs could act as predictive biomarkers to identify tumors whose cell growth is dependent on BET-mediated cMYC transcription. This finding shows the direct oncogenic functions of eRNAs and supports the possibility of targeting them for cancer intervention.

Another eRNA that also functions in prostate cancer is KLK3e. The current treatment for prostate cancer is androgen deprivation therapy, which mainly focuses on blocking the androgen signaling pathway [46,47,48,49]. KLK3e enhances the expression of AR-regulated genes in prostate tumors, providing the potential for eRNA-targeted therapy and the evaluation of androgen deprivation therapy in prostate cancer patients by measuring eRNA expression [13].

Zhang et al. observed the differential expression of eRNAs between tumor and control prostate tissues, which have the potential to target androgen response genes or act as prognostic biomarkers [31]. For example, an eRNA targets the SAM-pointed domain-containing Ets transcription factor gene, which acts as an androgen-independent activator of prostate-specific antigen promoters [31].

Similarly, several eRNAs are associated with bladder cancer tumorigenesis. Ding et al. detected up-regulated purinergic receptor P2RY2 enhancer RNA (P2RY2e) in bladder cancer tissues and estrogen-treated cells compared with controls [50], and showed that P2RY2e knockdown inhibited cell proliferation, invasion, and migration. Moreover, P2RY2e down-regulation reduced the cancer-promoting ability of estrogen on bladder cancer cells, suggesting that P2RY2e plays a carcinogenic role in bladder cancer [50].

Another example for bladder cancer is SMAD7e, an eRNA that is over-expressed in bladder cancer tissues and positively correlates with clinicopathological features. SMAD7e production is induced by estrogen, and its over-expression appears to promote the development of bladder cancer. SMAD7e knockdown led to cell apoptosis, and the suppression of cell proliferation, migration, and invasion, as well as an attenuation of the carcinogenic effect of estrogen in bladder cancer cells [51].

EBV is a common human herpesvirus associated with lymphoproliferative diseases such as Burkitt lymphoma and Hodgkin’s lymphoma [52,53]. Liang et al. found that EBV SE (ESE)-transcribed eRNAs are essential for the growth of lymphoblastoid cells, which are B-cell lines established by EBV infection [24,54]. MYC-428 eRNA and MYC-525 eRNA are transcribed from ESEs located –428 and –525 kb upstream of the MYC oncogene transcription site, respectively, and their knockdown by short hairpin RNA was shown to reduce MYC expression and lymphoblastoid cell growth. This suggests that ESE eRNA-targeted therapy could be used as a new treatment for EBV-associated malignancies. Similarly, the knockdown of AR eRNAs could inhibit or eliminate CRPC growth and open up a new avenue for cancer therapy. Table 1 summarizes previously mentioned eRNAs with potential to act as therapeutic targets in different cancers.

Besides this evidence from low-throughput molecular analyses, Zhang et al. also identified several clinically relevant eRNAs based on high-throughput genomic data analyses [27]. For example, TAO kinase (TAOK)1 eRNA, which targets TAOK1 in the Hippo signaling pathway, is associated with overall survival in kidney renal clear cell carcinoma. Additionally, engrailed (EN)1-associated eRNA targets the basal-like breast cancer marker gene EN1 and is highly expressed in this subtype. Similarly, CUGBP Elav-like family member 2 (CELF2)-associated eRNA, which putatively targets the tumor suppressor gene CELF246, is highly expressed in stomach adenocarcinoma [27].

Additionally, they explored 135 clinically actionable genes to determine whether there is a direct association between eRNAs and cancer therapy. Among these genes, the expression of 107 and 36 correlated with eRNAs in at least one and five cancer types, respectively. Additionally, the expression of several immune checkpoint proteins, such as B- and T-lymphocyte attenuator, programmed death-ligand (PDL)1, PDL2, hepatitis A virus cellular receptor 1, cluster of differentiation 200, and plectin, correlated with eRNAs in more than five cancer types. They also applied genomic data analyses to probe the role of eRNAs in the drug response. They identified 512 eRNAs that correlated with 63 anticancer drugs in 10 cancer signaling pathways. Of these, 217 showed a strong correlation with belinostat, a histone deacetylase inhibitor drug used for the treatment of hematological malignancies and solid tumors [57]. They further found that the putative target genes of 32.7% of these eRNAs are within the Notch signaling pathway, and that the remainder are in multiple pathways, including p53, cell cycle, and Wnt pathways. As these pathways are associated with cancer and cancer therapy, we can reasonably speculate that the eRNAs involved will affect the drug response. These results suggest the existence of interactions between eRNAs and clinically actionable genes or immune checkpoints, providing the potential for eRNAs to be targeted in cancer therapy [27].

Additionally, Guo et al. proved that the expression of eRNA AC003092.1 was significantly correlated to that of its target gene, and the survival in patients with glioblastoma multiforme [55]. Function enrichment analysis showed that AC003092.1-related genes were mainly over-represented in immune-associated categories and pathways. Additionally, the correlation between AC003092.1 and immune cell function is further proved by immunogenomic analysis, providing new insight into research on the immunosuppressive microenvironment of glioblastoma [55].

Gu et al. identified an eRNA AP001056.1 targeting an immune checkpoint protein, and it was also significantly associated with overall survival in squamous cell carcinoma of the head and neck. Given its role in immune response, AP001056.1 could serve as a prognostic marker [56].

## 5. Computational and Experimental Methods

eRNAs are unstable and degrade rapidly, so conventional detection methods that rely on steady-state RNA transcription are relatively inefficient and insensitive. Recently, new technologies have been developed for eRNA detection, such as global nuclear run-on sequencing (GRO-seq) and precision run-on sequencing (PRO-seq) and their derivatives, 5′GRO-seq and PRO-cap. GRO-seq is an exceptionally sensitive method of estimating transcriptional activity throughout the genome, which achieves its high sensitivity through multiple affinity purification steps [58]. PRO-seq is a genome-wide adaptation of nuclear run-on assays based on GRO-seq and is highly sensitive at detecting rare nascent RNAs with a large dynamic range [59]. 5′GRO-seq is a derivative of GRO-seq, which can identify 5′-capped nascent transcripts [60]. PRO-cap is a variant of PRO-seq that can also sequence the 5′-capped nascent RNA, and it can capture the synthesis level of nascent RNA [59]. These methods are all capable of measuring the transcription of enhancer regions (Table 2) [58,59,60,61,62,63].

Methods of identifying the association between eRNAs and cancer signaling pathways are worthy of attention. Zhang et al. established a correlation between eRNAs and signaling pathways by constructing a global eRNA–gene network across different cancer types [27]. They collected hundreds of genes associated with 10 cancer signaling pathways for correlation analysis. Examining the co-expression of putative target genes and eRNAs and limiting the distance between them to <1 Mb, they constructed a regulatory network with Spearman’s correlation Rs ≥ 0.3 and false discovery rates < 0.05. Hi-C data from control tissues were used to confirm the connections between eRNAs and their putative target genes [27]. Zhang et al. adopted a different strategy to define the target genes as those within 300 kb upstream or downstream of eRNAs. They used the R/Bioconductor package clusterProfiler and the Database for Annotation, Visualization and Integrated Discovery to perform the functional enrichment of target genes [31].

In addition to these computational methods used for eRNA function investigation, experimental methods are also worth mentioning. In order to investigate the potential functional roles of eRNAs in transcriptional regulation in cancers, RNA interference is commonly used. There are mainly two key approaches to induce RNA interference: double-stranded small interfering RNAs (siRNAs) and vector-based short hairpin RNAs (shRNAs). Additionally, siRNAs and shRNAs have been widely used for the investigation of eRNA functions in cancers and proved efficient. For example, Liang et al. used shRNA to assess the roles of MYC ESE eRNAs in LCL growth and survival, and shRNA knockdown reduced eRNAs expression levels by >80%. Hsieh et al. designed siRNAs targeting KLK3e on the basis of KLK3e mapping to investigate the regulatory function of KLK3e in prostate cancer. Li et al. designed specific siRNAs directed against eRNAs to investigate the potential roles of eRNAs on estrogen-dependent transcription activation events [13,20,24]. However, in some cases, in addition to the RNA interference method using specific siRNAs, antisense oligonucleotides against eRNAs were applied as well. For example, Li et al. used specific locked nucleic acid antisense oligonucleotides designed against eRNA transcripts to exclude the off-target effect of siRNAs. Likewise, Zhao et al. used highly optimized generation-2.5 antisense oligonucleotides that induce the degradation of the complementary target RNA without involving the cellular RNA interference machinery to avoid the potential off-target effects [20,30].

CRISPR-Cas13a is also an appropriate method for the targeted knockdown of eRNAs. Che et al. used CRISPR-Cas13a targeting SMAD7e to measure the effect of SMAD7e knockdown on biological behaviors of bladder cancer cells [51]. Ding et al. cultured and transfected bladder cells with specific CRISPR-Cas13a vectors and siRNA to investigate the influence of P2RY2e down-regulation on bladder cancer cells. Additionally, interestingly, they found the targeted knockdown of P2RY2e with CRISPR-Cas13a seemed much more efficacious, since siRNA failed to knock down P2RY2e in bladder cancer cells 5637 and T24, while the P2RY2e expression was decreased by 56.1% and 26.3% in 5637 cells and T24 cells by CRISPR-Cas13a (Table 2) [50].

## 6. Current eRNA Data Resouces

Since the important roles of eRNA in gene regulation and human cancers are being increasingly realized, different computational pipelines are developed to characterize the eRNA expression landscape across normal human tissues or cancer types by integrating multiple existing datasets. Additionally, based on these studies, several eRNA data resources are established.

Human enhancer RNA Atlas (HeRA) is a data resource that provides an expression landscape and regulatory network of eRNAs in normal human tissues by integrating multiple datasets such as ENCODE, FANTOM, and GTEx. HeRA is developed by Zhang et al., who collected annotations of enhancers from ENCODE, Ensembl 87, FANTOM, and Roadmap Epigenomics Project. Then, they integrated these annotations. More specifically, they extended 6kb around the middle site of enhancer as a potential eRNA region, and excluded those regions overlapping with known transcript regions annotated in at least one of Ensembl, GENCODE, UCSC databases. RNA-seq data collected from GTEx were mapped, and the expression level was calculated as reads per million (RPM). eRNAs with RPM ≥1 were considered as detectable eRNAs (Figure 3).

A great number of eRNAs are detected and associations between eRNAs and traits, including gender, race, and age are identified. HeRA allows users to explore, browse and download eRNA expression profiles, trait-related eRNAs, and putative regulators or target genes of each eRNA across different human tissues. Overall, HeRA is a valuable resource that facilitates functional and mechanistic investigations of eRNAs in normal activities of cells [64].

Unlike HeRA, eRNA in cancer (eRic) is a data resource developed for the users’ convenience to access and utilize data concerning eRNA in cancer. Zhang et al. integrated multi-omics and pharmacogenomics data across largescale patient samples and cancer cell lines to investigate the systemic landscape and potential function of eRNA in cancer and identified a panel of clinically relevant eRNAs. Additionally, based on these results, they built this data resource. eRic allows users to explore the expression of eRNA across different cancer types and samples in TCGA, and to identify the clinically relevant eRNAs. Furthermore, eRic can help to identify eRNA target genes, as well as to investigate individual eRNA’s sensitivity or resistance to drugs [27].

The Cancer eRNA Atlas (TCeA) is developed by Chen et al. to utilize the ultra-deep RNA-seq data in TCGA and GTEx for the exploration of eRNA expression patterns in super enhancers [65]. TCeA provides the annotation of eRNA loci in super enhancers. They first identified a set of regions with super enhancer activities in >20 (out of 86) tissue and cell types and defined them as core super enhancer regions (~5 Mb). They built a PCA model based on core regions, and further generalized their analysis to the whole set of super enhancer regions and identified over 300,000 precise eRNA loci located in 377 Mb regions of super enhancers, providing a high-resolution map of eRNA loci in super enhancers. Additionally, this high-resolution map makes it easier to measure super-enhancer activities in patient samples using routine RNA-seq data. Furthermore, TCeA has also quantified the activity of these eRNA loci in tens of thousands of tumor samples from TCGA, tissues samples from GTEx, and cell line samples from CCLE [65].

Taken together, these eRNA data resources benefit a broad range of biomedical investigations.

## 7. Conclusions and Perspectives

Identifying the critical role of eRNAs in regulating gene transcription and in the development of different cancer types has also enabled a better understanding of enhancers. Recent studies suggest that eRNAs regulate gene expression through promoting enhancer–promoter looping, and the knockdown of eRNAs is often accompanied by decreased target gene expression. Abundant evidence shows that eRNAs are abnormally expressed in various cancers and that their expression is closely related to tumorigenesis.

The expression of genes in signaling pathways correlates with eRNAs in various types of cancer. Thus, eRNA expression is regulated by signaling pathways, and, in turn, eRNAs also regulate certain signaling pathways. Although the associations between eRNAs and signaling pathways in human cancers are well-studied, the underlying molecular mechanisms involved remain unclear. Additionally, the relationship between eRNAs and signaling pathways has been mainly established through genomic research, with limited supporting evidence from molecular assays. Therefore, further molecular analyses are required to fully understand these mechanisms.

The characterization of eRNA functional roles in cancers may provide novel insights into cancer therapy. For example, the knockdown of specific eRNAs inhibits cell proliferation, invasion, and migration in cancer; thus, eRNA-targeted inhibitors could provide new entry points for cancer treatments. Additionally, the potential clinical value of eRNAs is suggested by the establishment of a correlation between eRNAs and clinically actionable genes and immune checkpoints. However, because cases that directly apply eRNAs to cancer treatment are limited, further efforts are required to fully explore the utility of eRNAs in cancer therapy.

At present, several eRNA data resources have been established, providing the research community with convenient and efficient platforms that facilitate the exploration for functional roles of eRNAs in both normal cells and cancer cells. With the boom of research on eRNA in the future and the growing number of samples coming along, these eRNA data resources may need to be updated continuously.

## Figures and Tables

**Figure 1 ijms-22-05640-f001:**
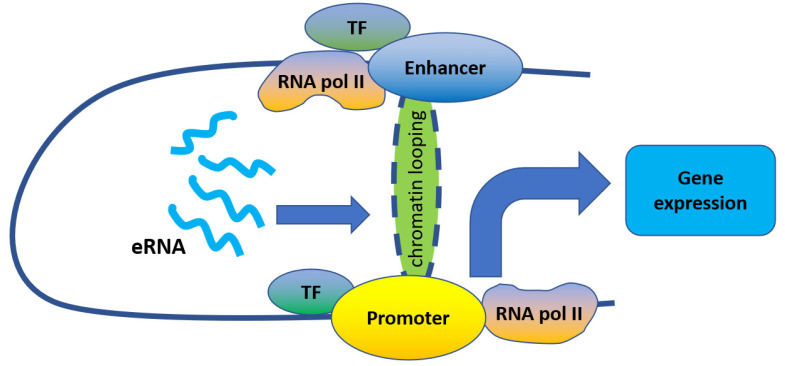
eRNAs regulate gene expression by promoting enhancer–promoter looping.

**Figure 2 ijms-22-05640-f002:**
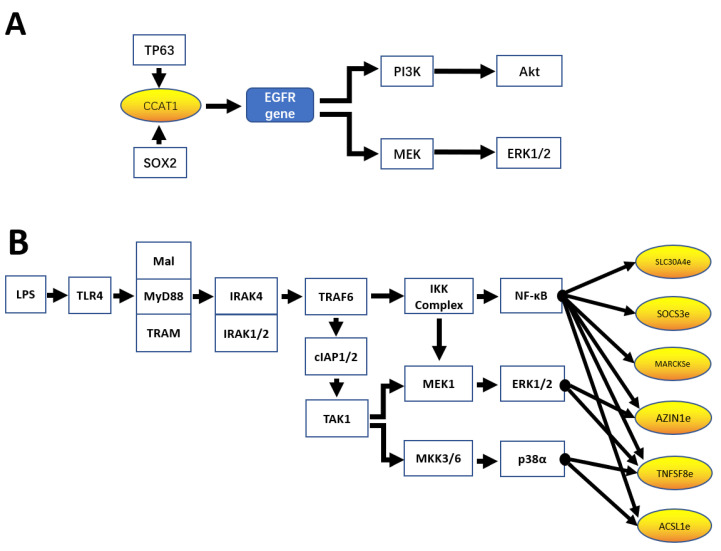
Pathways regulated by eRNA and pathways regulating eRNA expression. (**A**) eRNA CCAT1 activates PI3K/AKT and MEK/ERK1/2 signaling pathways. (**B**) Expression of LPS-induced eRNAs is regulated by divergent signaling pathways.

**Figure 3 ijms-22-05640-f003:**
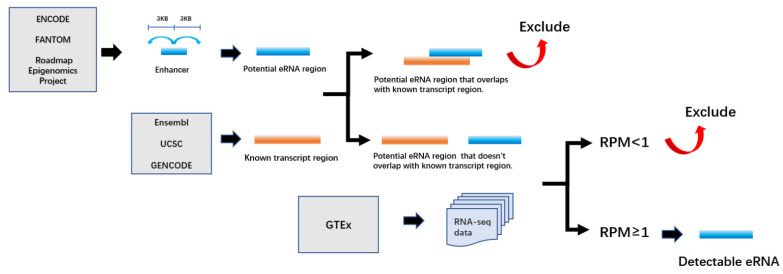
Pipeline for eRNA detection.

**Table 1 ijms-22-05640-t001:** eRNAs involved in different cancer types and the correlated signaling pathways.

eRNA	Cancer Type	Correlated Signaling Pathway	Reference
CCAT1	colon cancer; squamous cell carcinomas	PI3K/AKTK signaling; MEK/ERK signaling	Jiang et al. [19] McCleland et al. [44]
p53BER eRNA	p53-WT renal cancer	p53 signaling	Melo et al. [18]
PSA eRNA	prostate cancer	Androgen receptor signaling	Zhao et al. [30]
MYC ESE eRNAs	EBV-associated malignancies	MYC signaling	Liang et al. [24]
NET1e	breast cancer	/	Zhang et al. [27]
SMAD7e	bladder cancer	/	Che et al. [51]
P2RY2e	bladder cancer	/	Ding et al. [50]
AC003092.1	glioblastoma multiforme	/	Guo et al. [55]
AP001056.1	squamous cell carcinoma of the head and neck	/	Gu et al. [56]

**Table 2 ijms-22-05640-t002:** Methods for identification and functional investigations of eRNAs.

Methods	Description
Global Run-On sequencing (GRO-seq)	GRO-seq is able to assess nascent low-abundant RNAs genome-wide independent of the effect of RNA stability.
Precision Global Run-On sequencing (PRO-seq)	PRO-seq is based on GRO-seq, and highly sensitive and can identify short unstable nascent RNAs.
PRO-cap; 5′GRO-seq	PRO-cap and 5′GRO-seq are developed basing on PRO-seq and GRO-seq, and are able to identify the capped nascent RNA from 5′ end.
small interfering RNAs	The simplest method for RNA interference, siRNAs can inactivate their target RNAs in a sequence-specific way.
short hairpin RNAs	Another method of RNA interference. shRNAs can be transfected by delivery of plasmid vectors or through virally produced vectors.
CRISPR-Cas13a	As a novel type of RNA targeting enzyme, CRISPR-Cas13a is proved to be efficacious for RNA knockdown in mammalian cells.

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
