# Peer review of "Current Advances on the Important Roles of Enhancer RNAs in Molecular Pathways of Cancer"

_ijms, 2021, doi:10.3390/ijms22115640_

Round 1

Reviewer 1 Report

This review entitled "Current Advances on the Important Roles of Enhancer RNAs in Molecular Pathways of Cancer" by Wang and Tang is generally well-written and informative. However as an important review for the field, it is advised that the authors should include more figures and tables for the readers, in each section. 

Reviewer 2 Report

Wang et al review about “Current Advances on the Important Roles of Enhancer RNAs in 2 Molecular Pathways of Cancer” covers an interesting and relevant topic in the literature. Furthermore, the review is a good source of information on eRNAs in cancer.

I have some suggestions that can improve the review.

  • I suggest the author add some sentences about the characteristic of eRNAs, for their structure (size, spiced, polyadenylated, and capped) and how they are processed.
  • Line 122 remove “and” after Zhang and add the reference number.
  • The sentence that starts in Line 133 confuses, clarify the sentence.
  • Add refences in the following sentences, lines 132, 143, 154,1 77, 210, 215, 240,266, 288 and 350.
  • In item 6 (Current eRNA data resources) it will be interesting for the reader if the author adds 1 or 2 paragraphs about pipelines or articles that describe how to identify eRNAS.   
